# Brassinosteroids Benefit Plants Performance by Augmenting Arbuscular Mycorrhizal Symbiosis

Ying Ren,[a] Xianrong Che,[a] Jingwei Liang,[a] Sijia Wang,[a] Lina Han,[a] Ziyi Liu,[a] Hui Chen,[a] ⬤ Ming Tang[a]

[a]State Key Laboratory of Conservation and Utilization of Subtropical Agro-Bioresources, Guangdong Laboratory for Lingnan Modern Agriculture, Guangdong Key Laboratory for Innovative Development and Utilization of Forest Plant Germplasm, College of Forestry and Landscape Architecture, South China Agricultural University, Guangzhou, China

**ABSTRACT** Arbuscular mycorrhizal (AM) play an important role in improving plant growth and development. The interaction between phytohormones and AM symbiosis is gradually revealed. Here we examined the effect of Brassinosteroids (BR) on AM symbiosis and discussed the synergistic promotion of plant growth by BR and AM symbiosis. The xylophyta *Eucalyptus grandis* Hill (*E. grandis*) was inoculated with AM fungi *Rhizoglomus irregularis* R197198 (*R. irregularis*) and treated with different concentrations (0, 1, 10, and 100 nM) of 24-epibrassinolide (24-epiBL) for 6 weeks. With the increase of 24-epiBL concentration, *E. grandis* growth was firstly promoted and then inhibited, but inoculation with AM fungi alleviated this inhibition. 24-epiBL and *R. irregularis* colonization significantly improved *E. grandis* growth and antioxidant system response, and the synergistic effect was the best. Compared with the control group, 24-epiBL treatment significantly increased the mycorrhizal colonization and arbuscular abundance of AM fungi *R. irregular* in *E. grandis* roots. The expression of AM symbiosis maker genes was significantly increased by 24-epiBL treatment. Both 24-epiBL treatment and AM colonization upregulated gibberellins (GA) synthesis genes, but no inhibition caused by GA levels was found. 24-epiBL is a kind of synthetic highly active BR. Based on the results of 24-epiBL treatment, we hypothesized that BR actively regulates AM symbiosis regulates AM symbiosis without affecting *GA-INSENSITIVE DWARF1 (GID1)-DELLA* expression. The synergistic treatment of BR and AM symbiosis can significantly promote the growth and development of plants.

**IMPORTANCE** Brassinosteroids (BR) and Arbuscular mycorrhizas (AM) symbiosis play an important role in improving plant growth and development. Previous studies have shown that there is a complex regulatory network between phytohormones and AM symbiosis. However, the interactions of BR-signaling and AM symbiosis are still poorly understood. Our results suggest that BR actively regulates the colonization and development of AM fungi, and AM fungal colonization can alleviate the inhibition of plant growth caused by excessive BR. In addition, BR actively regulates AM symbiosis, but does not primarily mediate gibberellins-DELLA interaction. The synergistic treatment of BR and AM symbiosis can significantly promote the growth and development of plants. The conclusions of this study provide a reference for phytohormones–AM symbiosis interaction.

**KEYWORDS** Arbuscular mycorrhizal, Brassinosteroids, symbiosis, xylophyta

Arbuscular mycorrhizal (AM), a mutually beneficial endosymbiotic form of soil fungi and plants, is evolutionarily ancient and widely distributed (1). AM fungi transfer mineral nutrients (mainly phosphate) in the soil to plants to promote plant growth, and in exchange they obtain carbohydrates to maintain their own survival (2–4). After the AM fungal spores germinate in the soil, the hyphae continuously elongate and branch in search of a host (5). The root epidermis forms the pre-penetration apparatus (PPA) when it comes into contact with the hyphae. Then, the hyphae penetrate into the epidermal cells on this predefined path and eventually form arbuscular in the

**Ad Hoc Peer Reviewer** Hiromu Kameoka

Address correspondence to Hui Chen, chenhui@scau.edu.cn, or Ming Tang, tangm@nwafu.edu.cn.

endothelial cells (6). At the same time, the extraradical hyphae continued to develop. Along with the development of the arbuscular, the cell membrane expands to envelop it, thus forming the so-called periarbuscular membrane (PAM). The fungal plasma membrane, PAM, and the space between them constitute the symbiotic interface between fungi and plants. The symbiotic interface that avoids direct contact between fungal hyphae and plant cytoplasm ensures the signal perception and nutrient exchange between the two parties (7–9).

Phytohormones are known to be involved in almost all aspects of plant development. The regulatory network of various phytohormones and AM symbiosis is currently being uncovered. Strigolactones (SL) secreted from the host roots induces fungal hyphae branching and stimulates the fungus to secrete symbiotic signals (10). Increasing auxin content upregulates SL synthesis, thereby promoting AM symbiosis (11, 12). In addition, auxin is also involved in the drastic reorganization of microfilaments, microtubules, and cytoplasmic during the period when the hyphae penetrate the root cortex cells (12, 13). The gibberellins (GA) receptor GA INSENSITIVE DWARF1(GID1) interacts with the DELLA protein in a GA-dependent manner and triggers the 26s proteasome pathway to degrade the DELLA protein. The DELLA protein interacts with REDUCED ARBUSCULAR MYCORRHIZA1 (RAM1) to actively regulate multiple stages of arbuscular development (14). So GA negatively regulates AM symbiosis by suppressing DELLA (15).

Brassinosteroids (BR), known as the sixth largest phytohormone, contain more than 60 types of polyhydroxy steroid derivatives. The physiological role of BR involved in vascular differentiation, pollen development, and photomorphogenesis has been deeply studied (16). In addition, BR is also involved in the development of plant roots, such as cell elongation, meristem size, and lateral root development (17). In recent years, the interaction between BR and AM symbiosis is being revealed. The hyphal colonization of AM fungi *Rhizoglomus irregularis* R197198 (*R. irregularis*) in the BR-deficient *lkb* pea mutants or the wild-type plants did not have difference (18). However, the *lkb* mutants has been shown to still be able to synthesize one-fifth of the original content of BR (19). The AM colonization and arbuscular abundance of BR-deficient mutants in tomato (*dx*) and rice (*brd-1*) showed a significant decrease (20, 21). Based on these findings, we speculate that BR may have a positive effect on AM symbiosis. Here, *Eucalyptus grandis* Hill (*E. grandis*) plants were irrigated with different concentrations of 24-epibrassinolide (24-epiBL) for 6 weeks and either inoculated with AM fungi *R.irregularis* or uninoculated (AM or NM) (22, 23). Based on previous studies, we use woody plants as hosts for AM fungi. We have clarified the active regulation of BR on AM symbiosis and further proved that there may not be species-dependent differences.

## RESULTS

**24-epiBL application and AM fungi *R. irregularis* colonization promotes *E. grandis* growth.** We treated *E. grandis* with or without *R. irregularis* with 24-epiBL solutions (0, 1, 10, and 100 nM) for 6 weeks and then observed the plant phenotype. Compared with the untreated control group, the plants inoculated with *R. irregularis* or 24-epiBL treatment were more robust and taller (Fig. 1E).

We measured the roots fresh weight and tap roots length of AM versus NM *E. grandis* plants and saw that the AM *E. grandis* performed better under the same concentration of 24-epiBL treatment (Fig. 1A and C). With the increase of 24-epiBL concentration, the fresh roots weight and tap roots length increased first and then decreased gradually. The roots fresh weight and tap roots length of AM plants were the highest under 10 nM 24-epiBL treatment, which was significantly different from other treatment groups (Fig. 1A and C). Notably, the growth of *E. grandis* roots treated with 100 nM 24-epiBL was no different from that of the control group, but the addition of inoculation treatment still promoted the growth of roots. The shoots fresh weight and plant height of *E. grandis* gradually increased with the increase of 24-epiBL concentration, and that of AM plants were the highest under 10 and 100 nM 24-epiBL treatment, which was significantly different from other treatment groups (Fig. 1B and D). The shoots growth of AM plants was likewise better than that of NM plants (Fig. 1B and D).

To sum up, low concentration of BR and inoculation of AM fungi promoted the growth of *E. grandis* to some extent. Higher concentrations of BR may inhibit plant growth, but inoculation with AM fungi alleviates this inhibition.

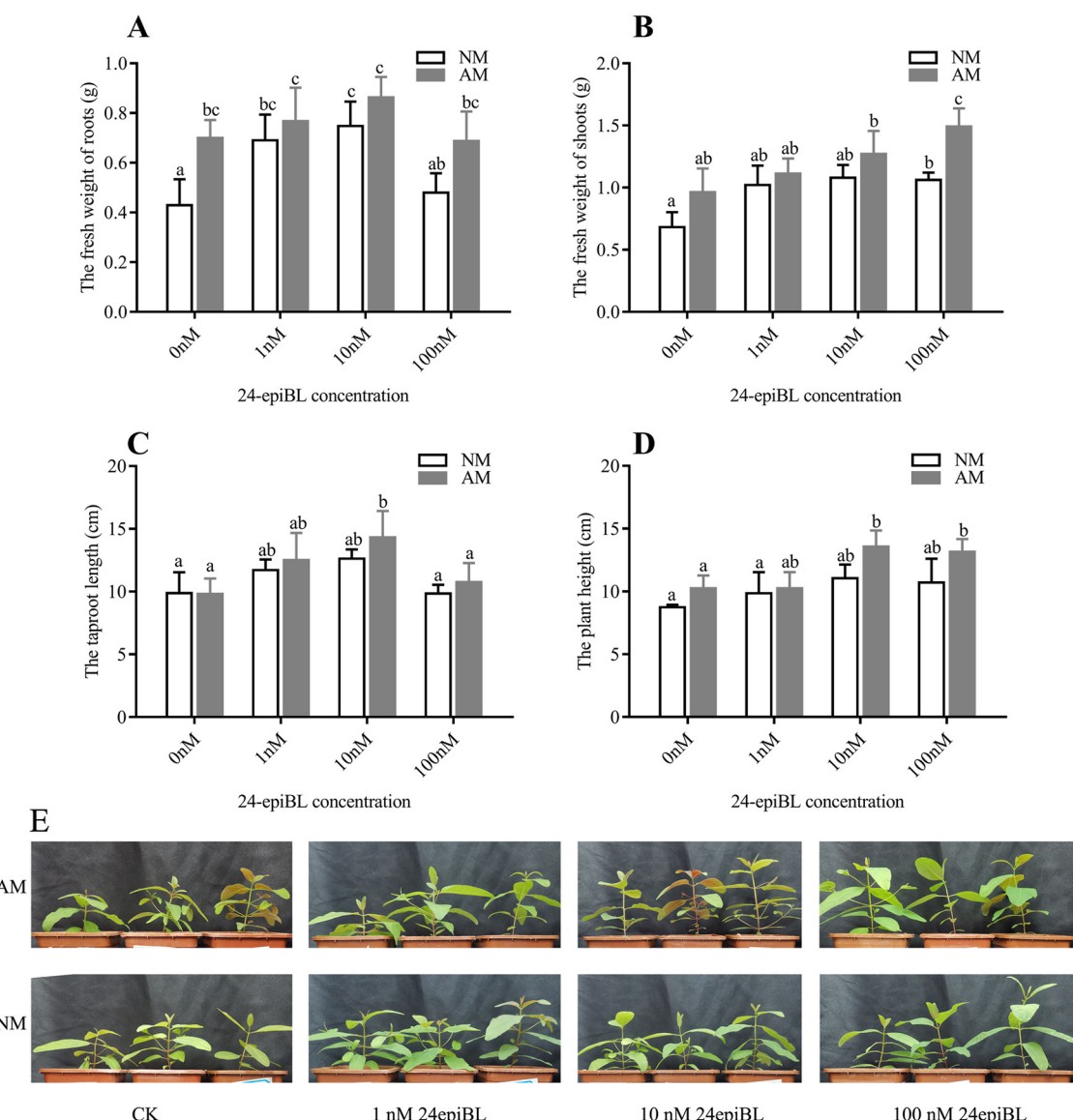

**FIG 1** Effects of AM fungi *R. irregularis* colonization and 24-epiBL application on the growth of *E. grandis*. 30-day-old *E. grandis* seedlings were inoculated (AM) and not inoculated (NM) with AM fungi *R. irregularis*, and then treated with different concentrations of 24-epiBL (1, 10 and 100 nM). After 6 weeks of treatment, the whole *E. grandis* plant sample was harvested, and the measurement results of its roots fresh weight (A), shoots fresh weight (B), tap roots length (C) and plant height (D). (E) Shoot growth performance of AM and NM *E. grandis* plants after 6 weeks of treatment. The data are shown as the means ± SE of six biological replicates ($n = 6$). Different letters indicate significant differences at $P < 0.05$, according to Duncan's new multiple range test.

**24-epiBL application promotes the mycorrhizal colonization of AM fungi *R. irregularis* in *E. grandis* roots.** Compared with the control group, 24-epiBL treatment significantly increased the mycorrhizal colonization frequency of AM fungi *R. irregularis* in *E. grandis* roots, and AM colonization frequency increased with the increase of 24-epiBL concentration (Fig. 2A). Similarly, as the concentration of 24-epiBL increased, the mycorrhizal intensity and arbuscular abundance also increased (Fig. 2B and C). The mycorrhizal intensity and arbuscular abundance of *E. grandis* roots under 10 and100 nM 24-epiBL were the highest and significantly different from other treatment groups, but the difference between them was not significant (Fig. 2B and C).

In a nutshell, the application of BR promoted the colonization of AM fungi *R. irregularis* in *E. grandis* roots and the development of AM arbuscular. However, given that the difference between 10 and 100 nM 24-epiBL was not significant, we speculated that excessively high concentrations of BR may no longer promote AM symbiosis.

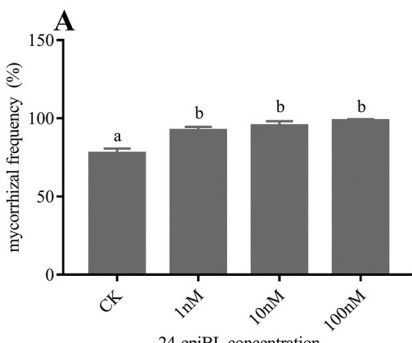

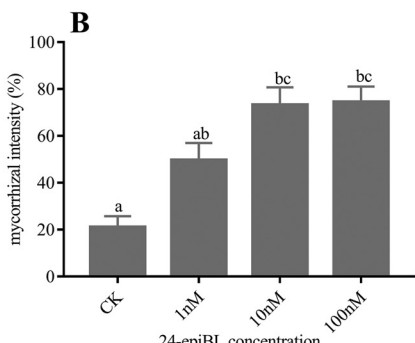

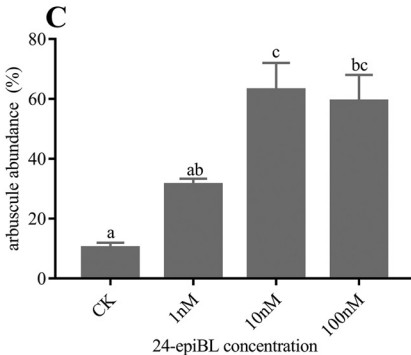

**FIG 2** Effects of application of different concentrations of 24-epiBL (1, 10 and 100 nM) on the colonization of AM fungi *R. irregularis* in *E. grandis* roots. Total mycorrhizal frequency (A), mycorrhizal intensity (B), and arbuscule abundance (C) in the *R. irregularis* colonized roots estimated after WGA488 staining. The data are shown as the means ± SE of three biological replicates (*n* = 3). Different letters indicate significant differences at *P* < 0.05, according to Duncan's new multiple range test.

**24-epiBL application and AM fungi *R. irregularis* colonization promotes *E. grandis* roots antioxidant system response.** Reactive oxygen species (ROS) are inevitably produced in the process of aerobic metabolism, and excessive ROS have toxic effects on plants. Antioxidant system can remove excess ROS to maintain the stability of ROS in plants. We determined the content of catalase (CAT), peroxidase (POD), superoxide dismutase (SOD), malondialdehyde (MDA), hydrogen peroxide ($H_2O_2$), and superoxide anion (OFR) in *E. grandis* roots treated with 24-epiBL and inoculated AM fungi. The content of CAT, POD, SOD reflected the activation degree of root antioxidant enzymes, MDA content reflected the degree of root lipid peroxidation damage, and the content of $H_2O_2$ and OFR reflected the level of active oxygen in the roots.

The activity of antioxidant enzymes (CAT, SOD, and POD) in *E. grandis* roots was enhanced under the application of 24-epiBL, and gradually increased with the increase of 24-epiBL concentration (Fig. 3A–C). On the contrary, the contents of MDA, $H_2O_2$, and OFR decreased with the increase of 24-epiBL concentration (Fig. 3D–F). The performance of antioxidant enzymes (CAT, SOD, and POD) activity in AM *E. grandis* plants roots was higher than that of NM plants (Fig. 3A–C). The contents of MDA, $H_2O_2$, and OFR in AM roots were lower (Fig. 3D–F). The AM *E. grandis* plants roots had the best antioxidant system response under 100 nM 24-epiBL treatment (Fig. 3).

These results indicated that both BR and AM fungi colonization can inhibit the accumulation of ROS and promote the activation of antioxidant enzymes, and the combination of the two treatments has the best effect.

**BR promotes the expression of AM symbiotic marker genes.** We also investigated the BR, AM, and GA-related genes expressions in AM *E. grandis* roots under 24-epiBL treatment to further explore the effect of BR on AM symbiosis (24).

BR binds to the extracellular domain of the BR receptor kinase BR-INSENSITIVE 1 (BRI1), which induces association and trans-phosphorylation between BRI1 and its co-receptor BR-INSENSITIVE 1-ASSOCIATED RECEPTOR KINASE 1 (BAK1), which in turn leads to

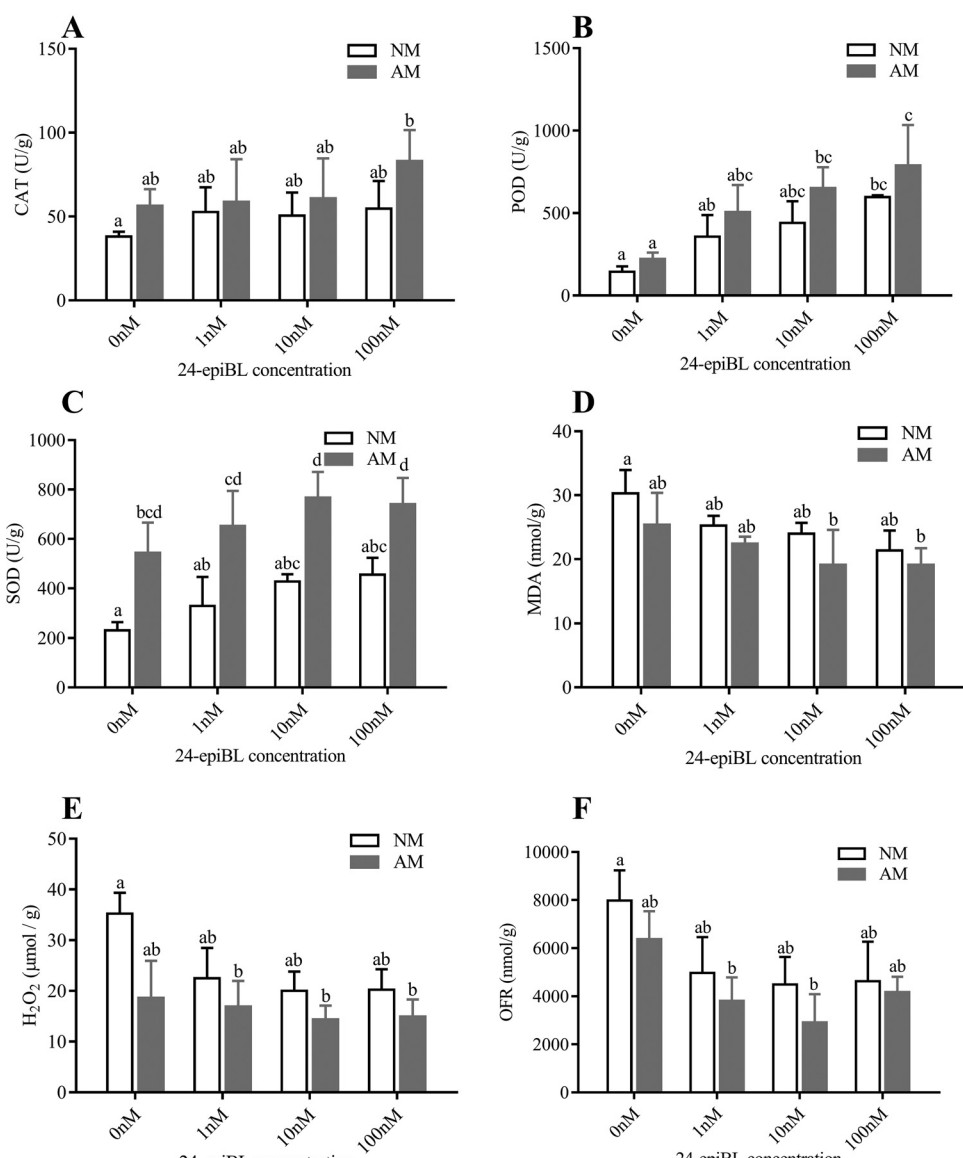

**FIG 3** Responses of *E. grandis* roots antioxidant system to AM fungi *R. irregularis* colonization and 24-epiBL application. Analysis of CAT (A), POD (B), SOD (C), MDA (D), H2O2 (E), and OFR (F) content in *R. irregularis* roots after 6 weeks of treatment (*R. irregularis* colonization and 24-epiBL application). The data are shown as the means ± SE of three biological replicates (*n* = 3). Different letters indicate significant differences at *P* < 0.05, according to Duncan's new multiple range test.

a phosphorylation cascade in the BR signaling pathway (25, 26). *EgBRI1* and *EgBAK1* were upregulated under 24-epiBL treatment, and their expression levels were significantly increased under the treatment of 10 and 100 nM 24-epiBL (Fig. 4A and B). CYP85A1 is a C-6 oxidase involved in BR biosynthesis (27). *EgCYP85A1* was downregulated under 24-epiBL treatment (Fig. 4C). In particular, AM fungal colonization upregulated the expression of BR receptor and synthetic genes, but there was no significant difference between AM and NM plants (Fig. 4C). Collectively, these results demonstrated that the application of 24-epiBL increased the BR level in *E. grandis* plants and affected the expression of BR signaling pathway genes.

GA 20-OXIDASE1 (GA20ox1) is an essential oxidase in gibberellin biosynthesis (28). GA signal is sensed by GID1 leading to the activation of GA downstream signaling pathways. DELLA proteins are negative regulator of the GA signaling pathway and interacts with GID1 that is regulated by the concentration of GA (29). In addition, DELLA proteins interact with RAM1 in the AM symbiotic pathway to promote arbuscular formation (15). RAM1 also actively regulates the

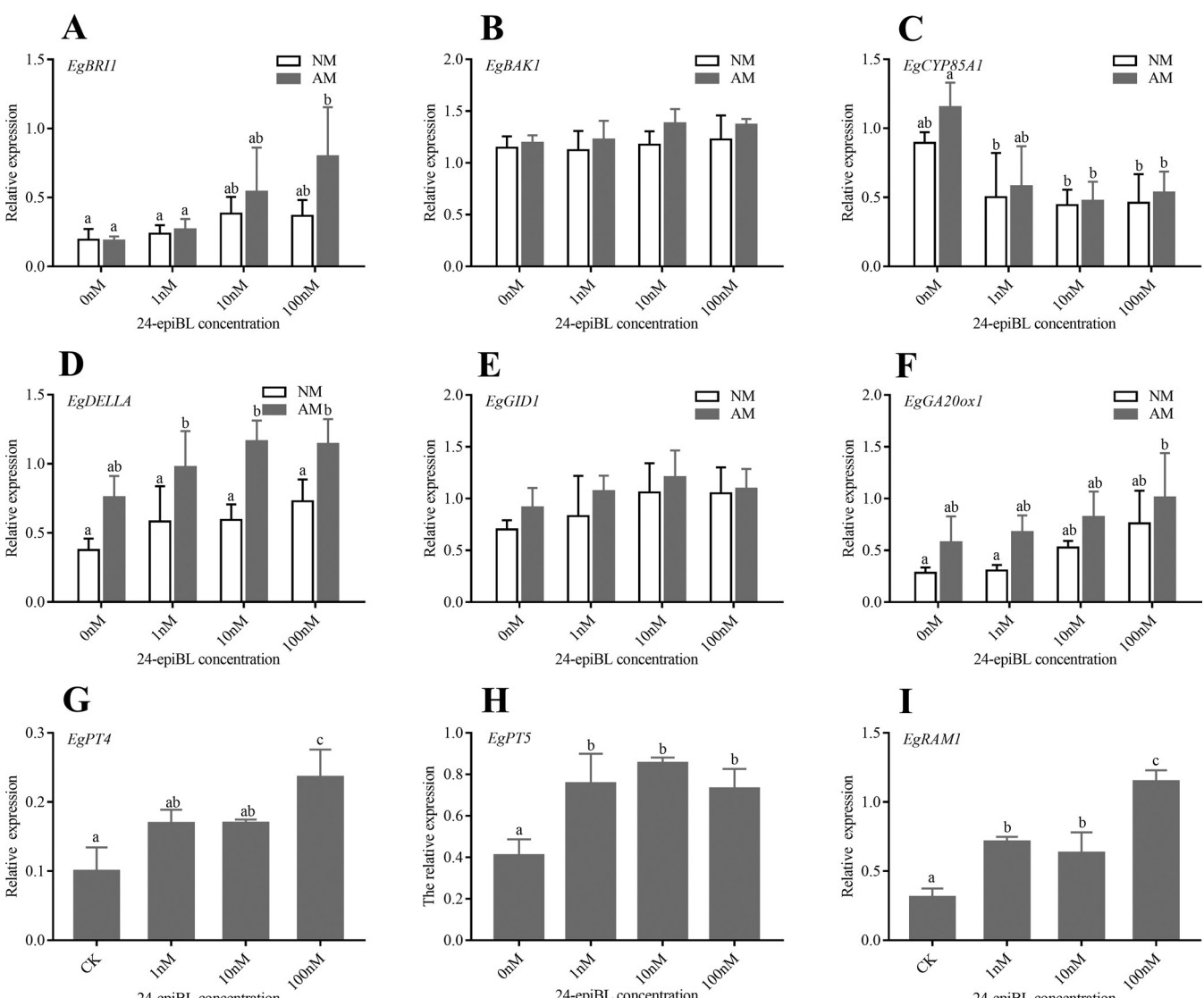

**FIG 4** Effects of 24-epiBL treatment on gene expression related to BR signaling, AM symbiosis and GA signaling in *E. grandis* roots. Total RNA was isolated from AM and NM *E. grandis* roots. The *EgUBI3* (Ubiquitin 3) gene from E. grandis was used as the housekeeping gene for normalization. Expression of the BR-related gene (*EgBRI1*, *EgBAK1* and *EgCYP85A1*) (A–C), the GA-related gene (*EgDELLA*, *EgGID1* and *EgGA20ox1*) (D–F), and the AM fungal-induced gene (*EgPT4*, *EgPT5* and *EgRAM1*) (G–I) in plant roots after 6 weeks of BR treatment. The data are shown as the means ± SE of three biological replicates (*n* = 3). Different letters indicate significant differences at *P* < 0.05, according to Duncan's new multiple range test.

penetration of AM fungal hyphae into root cortex cells (6). AM fungal colonization will specifically induce the expression of high-affinity phosphate transport genes in host plant roots (30).

We found that the expression of AM symbiosis maker genes (*EgRAM1*, *EgPT4*, and *EgPT5*) was significantly increased by 24-epiBL treatment in *R. irregularis*-inoculated *E. grandis* roots (Fig. 4G–I). And the expression of *EgDELLA* also showed a similar pattern (Fig. 4D). These results indicated that BR actively regulates AM symbiosis.

The expression of *EgGA20ox1* was significantly upregulated by AM fungal colonization (Fig. 4F). And the expression of *EgGA20ox1* increased with the increase of 24-epiBL concentration (Fig. 4F). The expression of *EgGID1* in the AM or BR treatment group was higher than that in the nontreatment group, but the difference was not significant (Fig. 4E). These results proved that both BR and AM fungi colonization can promote GA synthesis.

## DISCUSSION

**BR and AM symbiosis synergistically promote plant growth.** AM fungal colonization can promote the absorption and utilization of phosphate by host plants in a low-

phosphorus state (2). AM symbiosis enhance plant resistance to various abiotic/biological stresses by promoting plant development (31–33). BR plays an important role in plant development by promoting cell elongation and division, including the regulation of plant height, vascular differentiation, flower development, root development, and so on (34). In addition, BR alleviates oxidative stress and improves plant stress resistance (35, 36). These views are consistent with our research results. To a certain extent, BR and AM fungal colonization increased plant fresh weight, promoted the development of plant height and taproot length, inhibited the accumulation of ROS, and activated of antioxidant enzymes. In particular, excessive BR may inhibit plant growth, but AM fungal colonization can alleviate this inhibition. The growth status of plants combined with BR and AM fungal inoculation was the best. Together, these reports and results support that BR and AM symbiosis promote plant growth and development. In addition, our findings further suggest that BR and AM symbiosis have a synergistic effect on plant improvement.

**BR signal positively regulates AM symbiosis.** The pea (*Pisum sativum*) BR-deficient mutant *lkb* is a leaky mutation that can only synthesize 20% of the original BR level (37). The mycorrhizal colonization frequency and arbuscular development of AM fungal *Glomus intraradices* in *lkb* mutants was no different from that of the wild-type plants (18). This result means that the reduction of BR level has no effect on AM symbiosis. To further clarify whether this result is caused by the leaky nature of the mutation or because the BR level does not affect the AM symbiosis, they used the more severe BR-deficient mutant (*lk*) of pea (38). The result showed that the mycorrhizal colonization frequency and arbuscular abundance of AM fungal *Rhizobium leguminosarum* bv viciae (RLV248) in *lk* mutants were significantly lower than those in wild-type plants (39). This supports the hypothesis that BR influences AM symbiosis and indicates that the effect occurs on the basis of large changes in BR levels. Consistent with this view, the mycorrhizal colonization of AM fungi *R. irregularis* in tomato BR-deficient mutant (*dx*) and rice BR-deficient mutant (*brd-1*) was significantly reduced (40, 41). We used BR treatment to significantly increase the mycorrhizal colonization frequency, colonization intensity and arbuscular abundance of *R. irregularis* in *E. grandis* roots. Taken together, these data tell us that BR level promotes the colonization of AM fungi and positively regulates AM symbiosis.

Although the effect of BR on AM colonization is significant, the joint effects of BR and AM on plant growth can enhance plant growth under certain conditions. It is possible that the impact on plant performance would be greater under other environmental conditions and pressures, but this needs further experiments.

**BR regulates AM symbiosis without affecting GID1-DELLA expression.** Ethylene (ET) is a negative regulator of AM symbiosis, and the mycorrhizal colonization of the pea ET-defective mutant (*ein2*) was significantly enhanced (39). However, the mycorrhizal colonization of the *lk ein2* double mutant was not different from that of the *lk* single mutant, and both were significantly lower than the wild-type plants (39). This suggests that BR regulates AM symbiosis through another direct pathway rather than by mediating ethylene levels.

The sucrose transporter 2 of tomato (SlSUT2) participates in the recovery of sucrose from the symbiosis interface to the plant cytoplasm. The mycorrhizal colonization of SlSUT2-antisense tomato plants had increased significantly. Therefore, SlSUT2 negatively regulates AM symbiosis by limiting the supply of carbon sources (40). It is important to note that BR treatment can partially rescue the phenotype caused by SlSUT2 downregulation (40). BR treatment restored the phenotype of SlSUT2 silent tomato plants (low pollen vigor and germination rate, dwarfing) to a certain extent. These results suggest an association between BR signaling pathway and SlSUT2. And then three SlSUT2-interacting proteins were screened in BR signaling pathway, including the sterol reductase DWARF1/DIMINUTO1 (DIM1), the membrane-steroid binding protein 1 (MSBP1) and BAK1 (41). Taken together, these reports suggest that BR regulates AM symbiosis by mediating carbon source transport pathways. In addition, the interaction between BAK1 and MSBP1 has also been discovered. MSBP1 can act as a negative regulator of BR signaling (41). Therefore, BR actively regulates AM symbiosis, which may be hindered by MSBP1.

The interaction between BR and GA is both in synthesis and signaling. The GID1-GA complex formed by the combination of GID1 and GA can interact with DELLA protein and make it degraded by ubiquitination (42). The DELLA protein interacts with BR RESISTANT 1 (BZR1), a positive regulator downstream of the BR signaling pathway, interferes its function. BZR1, a downstream regulator of BR signaling, can bind to the promoter of GA synthesis genes *GA 20-oxidase* (*GA20ox*) and *GA 3-oxidase* (*GA3ox*) to induce their expression (14, 43). The DELLA protein, as a repressor of GA signaling, interferes with BR signal transduction. And the DELLA-BZR1 interaction regulates GA synthesis. Our results showed that BR treatment upregulated the expression of *GA20ox1*. Together, BR can promote GA synthesis.

It is intriguing that the DELLA protein participates in arbuscular formation by enhancing RAM1 activity (8). Consistent with this, *DELLA* expression in AM plants was significantly higher than that in NM plants in our study. Several studies have shown GA inhibits arbuscular formation and is a negative regulator of AM symbiosis (15, 44). In particular, it has also been reported that the signaling activity in the early stage of AM symbiosis stimulates some genes in GA signaling pathway (both positive and negative regulators). There may be a steady-state local regulation system in the symbiosis of GA and AM. In the early stage of AM symbiosis, the hyphae enters the host cell to induce the upregulation of GA synthesis around AM fungi to prevent repeated infection at the same invasion point and improve the invasion efficiency. The high concentration of GA accumulated in the late stage of AM symbiosis inhibits the colonization of new fungi to prevent over-infection of the host plant (44). Our results suggest that AM colonization upregulated the expression of *GA20ox1*. However, BR treatment did not affect *DELLA* and *GID1* expression levels. Meanwhile, BR treatment significantly increased the expression levels of AM symbiotic marker genes (*EgPT4*, *EgPT5* and *EgRAM1*). Taken together, we hypothesized that BR promotes AM symbiosis not by regulating the expression of *GID1*and *DELLA*. Whether BR mediates GA-signaling pathway to affect AM symbiosis needs to be further explored by relying on GA-related mutants.

**Conclusion.** BR and AM symbiosis play an important role in improving plant growth and development. Our results suggest that BR actively regulates the formation and development of AM symbiosis. And the synergistic treatment of BR and AM colonization has quite a small effect on plant performance, although the effect of BR on AM colonization was much more significant. Moreover, it is possible that their effects on plant performance may be greater under other environmental conditions and stresses, but additional studies are needed to prove this. Interestingly, BR promotes plant development not only through its own signaling pathway, but also through an indirect pathway that promotes AM symbiosis. There is a complex regulatory network between phytohormones and AM symbiosis. We know too little regarding the molecular modes of BR and AM symbiosis. By studying phytohormones interactions and microbial-plant interactions, combined with the use of single-gene and two-gene mutants, we can gain insight into cross talk and regulation of these signaling pathways.

## MATERIALS AND METHODS

**Plant culture, BR application, and AM fungus inoculation.** The *Eucalyptus grandis* Hill (*E. grandis*) seeds were surface-disinfected with 3% sodium hypochlorite for 20 min and rinsed with sterile distilled water. The seeds were germinated on a 1/4 MS medium at 25°C in the dark until the hypocotyls elongated about 1 cm, and then transferred to the growth chamber with a cycle of 16 h light (26°C) and 8 h darkness (22°C).

After 4 weeks of cultivation, the seedlings were transplanted into small plastic pots (8 × 8 × 8 cm) filled with quartz sand and inoculated with AM fungi *Rhizoglomus irregularis* R197198 (*R. irregularis*). Quartz sand was sterilized at 121°C for 2 h with an autoclave before use. Spores of *R. irregularis* germinated in sterile distilled water for 5 d at 25°C in the dark in advance. The inoculation was carried out by pipetting 1 ml of an aqueous solution containing about 500 spores to the vicinity of the root system. The 24-epiBL powder was dissolved in a small amount of anhydrous ethanol, then diluted to 20 mM in sterile water and stored at −20°C. The plants were watered with the improved Long Ashton nutrient solution containing 30 $\mu$M the standard phosphate concentration (22). 20 mM 24-epiBL was diluted with nutrient solution to 1, 100, and 100 nM, respectively, to BR treatment. AM fungal inoculation × BR two-factor randomized block design was used, with six plant biological replicates per treatment. The plant samples were stored at −80°C for the determination of various indicators.

**Evaluation of mycorrhizal colonization degree.** Part of the fresh roots of each AM plant were randomly selected and immersed in 10% KOH (wt/vol) at 90°C for 8 h (replaced with fresh KOH solution every 2 h), neutralized with 2% HCL (wt/vol) for 20 min, washed three times with sterile water, and then stained with 5 mM wheat germ agglutinin Alexa Fluor 488 (WGA488; Invitrogen, Carlsbad, CA) according to the manufacturer's instructions. Fluorescent signals in the stained fungal tissues were examined using a fluorescence microscopy (Nikon Y-TV55). And the colonization frequency and arbuscular abundance were calculated by the MYCOCALC program (http://www2.dijon.inra.fr/mychintec/Mycocalc-prg/download.html) (23).

**Analysis of antioxidant stress.** About 1 g of fresh plant tissue samples (or 0.01–0.05 g freeze-dried samples) were weighed, added 1 ml extract (selected the corresponding extract according to manufacturer's instructions) for ice bath homogenization, then centrifuged 10,000 g at 4°C for 20 min, took the supernatant and placed the supernatant on ice to be tested. And then the determination of antioxidant enzyme activity, MDA, OFR, and $H_2O_2$ content were performed with CAT Kit, SOD Kit, POD Kit, MDA Kit, OFR Kit, and $H_2O_2$ Kit (Solarbio, Beijing, China) according to the manufacturer's protocol.

**Gene expression.** Total RNA was isolated by the CTAB-LiCl method (24). cDNA was synthesized from 1 μg of RNA with ChamQ Universal SYBR qPCR Master Mix (Vazyme, Nanjing, China) according to the manufacturer's instructions. Real-time PCRs were performed using HiScript III RT SuperMix for qPCR (+gDNA wiper) (Vazyme, Nanjing, China) according to the manufacturer's instructions. Primers were designed to obtain a 50–150-bp amplicon using the https://sg.idtdna.com/Scitools/Applications/RealTimePCR/ online website. The primer sequences are provided in Table S1 in the supplemental material.

**Statistical analysis.** All data were analyzed using SPSS 16 software (SPSS, Chicago, IL) for two-factor analysis of variance (ANOVA) and $t$-tests. The data were presented as the mean $\pm$ SE of different replicates. Different letters indicate a significant difference at $P < 0.05$.

## SUPPLEMENTAL MATERIAL

Supplemental material is available online only.

**Supplemental File 1**, XLSX file, 0.01 MB.

## ACKNOWLEDGMENTS

This work was supported by the National Natural Science Foundation of China [Grant number: 32071639], by the Laboratory of Lingnan Modern Agriculture Project [Grant number: NZ2021025], and by the Science and Technology Planning Project of Guangdong Province [Grant number: 201904020022].

We declare no conflicts of interest.

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
