## [Reviewer comments · Microbiology Spectrum]

Microbiology Spectrum

Brassinosteroids benefit plants performance by augmenting arbuscular mycorrhizal symbiosis

Ying Ren, Xianrong Che, Jingwei Liang, Sijia Wang, Lina Han, Ziyi Liu, Hui Chen, and Ming Tang

Corresponding Author(s): Ming Tang, South China Agricultural University

Review Timeline:

Submission Date:	September 22, 2021
Editorial Decision:	October 12, 2021
Revision Received:	November 1, 2021
Editorial Decision:	November 2, 2021
Revision Received:	November 5, 2021
Accepted:	November 8, 2021

Editor: Lindsey Burbank

Reviewer(s): Disclosure of reviewer identity is with reference to reviewer comments included in decision letter(s). The following individuals involved in review of your submission have agreed to reveal their identity: Hiromu Kameoka (Reviewer #2)

Transaction Report:

DOI: <https://doi.org/10.1128/spectrum.01645-21>

October 12, 2021

Prof. Ming Tang
South China Agricultural University
College of Forestry and Landscape Architecture
No.483, Wushan Road, Tianhe District
Guangzhou
China

Re: Spectrum01645-21 (Brassinosteroids promote plant performance by augmenting arbuscular mycorrhizal symbiosis)

Dear Prof. Ming Tang:

Thank you for submitting your manuscript to Microbiology Spectrum. The topic of your study is interesting and relevant to the broad field of microbiology served by Microbiology Spectrum. However, the reviewers gave some critical feedback for your manuscript which needs to be addressed. Please carefully consider all the comments of the reviewers and modify your manuscript accordingly. Note that Microbiology Spectrum accepts studies with negative findings and incremental advances as these types of work are also crucial.

When submitting the revised version of your paper, please provide (1) point-by-point responses to the issues raised by the reviewers as file type "Response to Reviewers," not in your cover letter, and (2) a PDF file that indicates the changes from the original submission (by highlighting or underlining the changes) as file type "Marked Up Manuscript - For Review Only". Please use this link to submit your revised manuscript - we strongly recommend that you submit your paper within the next 60 days or reach out to me. Detailed information on submitting your revised paper are below.

Link Not Available

Sincerely,

Lindsey Burbank

Journals Department
Reviewer comments:

Reviewer #1 (Comments for the Author):

Line 25

Please write 24-epiBL also in full

Line 32-33

Authors should not generalize findings for a brassinosteroid to all brassinosteroids

Line 53

Please revise; this section is in the past tense

Line 88

Material and method -> Materials and methods

Line 105

Revise

Line 109

More details are needed; sample preparation? Sample volumes?

Line 118

Primer sequences can be moved to supplementary material

Figure 4J - I believe this figure was not refereed to in the text

Reviewer #2 (Comments for the Author):

1. The authors lead some conclusions depending on the date that are not statistically significant. For example, the title says "Brassinosteroids promote plant performance by augmenting arbuscular mycorrhizal"; however, there is no statistically significant differences in the fresh weight of roots, the taproot length, and the plant height between AM-inoculated and uninoculated plants treated with the same concentration of 24-epiBL (Fig. 1A, C, D). Although the fresh weight of shoots of AM-inoculated plants was increased compared to uninoculated plants only when 100 nM of 24-epiBL was treated, it was not statistically significantly increased when 10 nM of 24-epiBL was treated, which induced the same level of AM colonization as 100 nM of 24-epiBL treatment (Fig. 1B, Fig. 2). Similarly, many discussions about Figs. 1, 3, and 4 were based on statistically non-significant data.
2. The authors argue that "both BR and AM fungi colonization can promote the response of the plant's antioxidant system and alleviate oxidative stress, and the combination of the two treatments has the best effect" (lines 177-170), based on the results that AM inoculation and BL treatment increased the antioxidant enzyme activities (Fig. 3). However, it is possible that these enzymes were activated as a result of increased ROS levels by AM inoculation and BL treatment. Therefore, it is unclear whether AM inoculation and BL treatment are good for plants to prevent oxidative stress.
3. The authors argue that "BR mainly regulates AM symbiosis through GA-independent pathways" (lines 242), based on the results that BL treatment did not affect the expression levels of DELLA and GID1 (Fig. 3). However, it is possible that BL regulates AM symbiosis through GA related factors other than the expression levels of DELLA and GID1. To conclude the GA-independent BL functions, other experiments such as BL treatment on GA-related mutants are required.

Staff Comments:

Preparing Revision Guidelines

Please return the manuscript within 60 days; if you cannot complete the modification within this time period, please contact me. If you do not wish to modify the manuscript and prefer to submit it to another journal, please notify me of your decision immediately so that the manuscript may be formally withdrawn from consideration by Microbiology Spectrum.

Dear editors and reviewers,

Thank you for your thorough review of our work and we are very grateful for your input. We have revised our manuscript thoroughly based on your comments. Details are as follows:

Reviewer #1

Comment 1: Line 25, Please write 24-epiBL also in full.

Response 1: Thanks for the reviewer's kind suggestion. We have changed "24-epiBL" to "24-epibrassinolide (24-epiBL)" according to your comment. The revised details can be found in **Line 25**.

Comment 2: Line 32-33, Authors should not generalize findings for a brassinosteroid to all brassinosteroids.

Response 2: Thanks for the reviewer's kind suggestion. As a kind of synthetic BR analogue, 24-epiBL is feasible to explore the physiological effects of BR in plants. However, there is no denying that its activity is not exactly the same as the various BRs found in plants. Your suggestion is very good, so we have revised our expressions in the text. I can revise it again if you think there is a more accurate statement. The revised details are as follows:

Line 32-34, 24-epiBL is a kind of synthetic highly active BR. Based on the results of 24-epiBL treatment, we hypothesized that BR actively regulates AM symbiosis regulates AM symbiosis without affecting *GIDI-DELLA* expression.

Comment 3: Line 53, Please revise; this section is in the past tense.

Response 3: Thanks for the reviewer's kind suggestion. We have corrected "After the AM fungal spores germinated in the soil, the hyphae continuously elongated and branched in search of a host" to "After the AM fungal spores germinate in the soil, the hyphae continuously elongate and branch in search of a host" according to your comment. The revised details can be found in **Line 53-54**.

Comment 4: Line 88, Material and method -> Materials and methods.

Response 4: Thanks for the reviewer's kind suggestion. We have corrected "Material and method" to

“Materials and methods” according your comment. The revised details can be found in **Line 89**.

Comment 5: Line 105, Revise

Response 5: Thanks for the reviewer’s kind suggestion. According to his/her advices, we have added the operation details of the root pre-processing and rewritten the entire paragraph. The revised details are as follows:

Line 107-113, Part of the fresh roots of each AM plant were randomly selected and immersed in 10% KOH (w /v) at 90°C for 8 h (replaced with fresh KOH solution every 2 h), neutralized with 2% HCL (w /v) for 20 min, washed 3 times with sterile water, and then stained with 5 mM wheat germ agglutinin Alexa Fluor 488 (WGA488; Invitrogen, Carlsbad, CA, USA) according to the manufacturer’s instructions. Fluorescent signals in the stained fungal tissues were examined using a fluorescence microscopy (Nikon Y-TV55). And the colonization frequency and arbuscular abundance were calculated by the MYCOCALC program (<http://www2.dijon.inra.fr/mychintec/Mycocalc-prg/download.html>) (23).

Comment 6: Line 109, More details are needed; sample preparation? Sample volumes?

Response 6: Thanks for the reviewer’s kind suggestion. According to his/her advices, we have added more information about methods. The revised details are as follows:

Line 104-105, The plant samples were stored at -80°C for the determination of various indicators.

Line 115-120, About 1g of fresh plant tissue samples (or 0.01-0.05g freeze-dried samples) were weighed, added 1ml extract (selected the corresponding extract according to manufacturer's instructions) for ice bath homogenization, then centrifuged 10000g at 4°C for 20min, took the supernatant and placed the supernatant on ice to be tested. And then the determination of antioxidant enzyme activity, MDA, OFR and H₂O₂ content were performed with CAT Kit, SOD Kit, POD Kit, MDA Kit, OFR Kit, and H₂O₂ Kit (Solarbio, Beijing, China) according to the manufacturer's protocol.

Comment 7: Line 118, Primer sequences can be moved to supplementary material.

Response 7: Thanks for the reviewer’s kind suggestion. According to his/her advices, we have changed the

primer sequence in the main manuscript to supplementary material and uploaded it. The revised details can be found in **Line 127-128**.

Comment 8: Figure 4J - I believe this figure was not refereed to in the text.

Response 8: Thanks for the reviewer's kind suggestion. "Fig. 4G, H&I" in the text actually refers to "Fig. 4H, I&J" marked on the image respectively, and we have corrected the numbers on the image. The details related to it can be found in **Line208**, and **Line434**.

Reviewer #2 :

Comment 1: The authors lead some conclusions depending on the data that are not statistically significant.

For example, the title says "Brassinosteroids promote plant performance by augmenting arbuscular mycorrhizal"; however, there is no statistically significant differences in the fresh weight of roots, the taproot length, and the plant height between AM-inoculated and uninoculated plants treated with the same concentration of 24-epiBL (Fig. 1A, C, D). Although the fresh weight of shoots of AM-inoculated plants was increased compared to uninoculated plants only when 100 nM of 24-epiBL was treated, it was not statistically significantly increased when 10 nM of 24-epiBL was treated, which induced the same level of AM colonization as 100 nM of 24-epiBL treatment (Fig. 1B, Fig. 2). Similarly, many discussions about Figs.1, 3, and 4 were based on statistically non-significant data.

Response 1: Thanks for the reviewer's kind suggestion. As stated in the comments, the differences between AM plants, different concentrations 24-epiBL plants and the control group are not all statistically significant. However, AM plants and BR-treated plants performed better than the NM group. Even if they are not statistically significant, it can also indicate that AM and BR have an insignificant promoting effect on plant growth. Therefore, according to the suggestion, we have revised the title to more accurately reflect the results of our experiment. If there is a more accurate statement, we can modify it again. The revised details are as follows:

Line 1, Brassinosteroids benefit plants performance by augmenting arbuscular mycorrhizal symbiosis

Comment 2: The authors argue that "both BR and AM fungi colonization can promote the response of the plant's antioxidant system and alleviate oxidative stress, and the combination of the two treatments has the best effect" (lines 177-170), based on the results that AM inoculation and BL treatment increased the antioxidant enzyme activities (Fig. 3). However, it is possible that these enzymes were activated as a result of increased ROS levels by AM inoculation and BL treatment. Therefore, it is unclear whether AM inoculation and BL treatment are good for plants to prevent oxidative stress.

Response 2: Thanks for the reviewer's kind suggestion. According to his/her advices, we have supplemented the determination of hydrogen peroxide (H_2O_2) and superoxide anion (OFR) as representative indicators of roots ROS level. The results showed that AM colonization and 24-epiBL treatment could inhibit the accumulation of H_2O_2 and OFR in roots. Combined with the above results of antioxidant enzymes and MDA, we speculated that AM colonization and BR treatment were beneficial to prevent oxidative stress by activating antioxidant enzymes and inhibiting ROS accumulation. However, how BR-signaling and AM symbiotic-signaling activate plant antioxidant responses need to be further explored. The revised details are as follows:

Line 169-174, We determined the content of catalase (CAT), peroxidase (POD), superoxide dismutase (SOD), malondialdehyde (MDA), hydrogen peroxide (H_2O_2), and superoxide anion (OFR) in *E. grandis* roots treated with 24-epiBL and inoculated AM fungi. The content of CAT, POD, SOD reflected the activation degree of root antioxidant enzymes, MDA content reflected the degree of root lipid peroxidation damage, and the content of H_2O_2 and OFR reflected the level of active oxygen in the roots.

Line 177-180, On the contrary, the contents of MDA, H_2O_2 , and OFR decreased with the increase of 24-epiBL concentration (Fig. 3 D, E&F). The performance of antioxidant enzyme (CAT, SOD and POD) activity in AM *E. grandis* plants roots was higher than that of NM plants (Fig. 3 A, B&C). The contents

of MDA, H₂O₂ and OFR in AM roots were lower (Fig. 3 D, E&F).

Line 182-184, These results indicated that both BR and AM fungi colonization can inhibit the accumulation of reactive oxygen species and promote the activation of antioxidant enzymes, and the combination of the two treatments has the best effect.

Line 222-224, BR and AM fungal colonization increased plant fresh weight, promoted the development of plant height and taproot length, inhibited the accumulation of reactive oxygen species, and activated of antioxidant enzymes.

Line 425, Analysis of CAT (A), POD (B), SOD (C), MDA (D), H₂O₂ (E), and OFR (F) content in *R. irregularis* roots after 6 weeks of treatment (*R. irregularis* colonization and 24-epiBL application).

Comment 3: The authors argue that "BR mainly regulates AM symbiosis through GA-independent pathways" (lines 242), based on the results that BL treatment did not affect the expression levels of DELLA and GID1 (Fig. 3). However, it is possible that BL regulates AM symbiosis through GA related factors other than the expression levels of DELLA and GID1. To conclude the GA-independent BL functions, other experiments such as BL treatment on GA-related mutants are required.

Response 3: Thanks for the reviewer's kind suggestion. According to his/her advices, we have revised our presentation to reflect our results more clearly. In addition, it is our further research content to explore whether BR-signaling influences AM symbiosis to mediate GA pathway. This suggestion is of great importance to us. We will refine our next step by BL treatment on GA-related mutants. The revised details are as follows:

Line 185, BR promotes the expression of AM symbiotic marker genes

Line 245, BR regulates AM symbiosis without affecting *GID1-DELLA* expression

Line 285-288, Taken together, we hypothesized that BR promotes AM symbiosis not by regulating the expression of *GID1* and *DELLA*. Whether BR mediates GA-signaling pathway to affect AM symbiosis needs to be further explored by relying on GA-related mutants.

All the changes we make in the manuscript are highlighted by coloured text.

If there are any further issues, please don't hesitate to tell us.

Best regards!

Yours

Ming Tang

November 2, 2021

Prof. Ming Tang
South China Agricultural University
College of Forestry and Landscape Architecture
No.483, Wushan Road, Tianhe District
Guangzhou
China

Re: Spectrum01645-21R1 (Brassinosteroids benefit plants performance by augmenting arbuscular mycorrhizal symbiosis)

Dear Prof. Ming Tang:

Thank you for submitting your manuscript to Microbiology Spectrum. As you will see your paper is very close to acceptance. Thank you for carefully considering the comments of the reviewers and providing additional information. In my assessment of your responses there is just one more point that I would like to see addressed: similar to the points of reviewer #2, I think it is important to make clear that the effect of BR+AM on plant performance in general was quite small although the effect of BR on AM colonization was much more significant. As this was an experimental condition, it is possible that the impact on plant performance would be greater under other environmental conditions and pressures, but that would be an area for additional work. If you could address this point specifically in the conclusions I think it would be beneficial for those who are interested in this area of research and want to build on your results.

Please modify the manuscript along the lines I have recommended. As these revisions are quite minor, I expect that you should be able to turn in the revised paper in less than 30 days, if not sooner.

When submitting the revised version of your paper, please provide (1) point-by-point responses to the issues I raised in your cover letter, and (2) a PDF file that indicates the changes from the original submission (by highlighting or underlining the changes) as file type "Marked Up Manuscript - For Review Only". Please use this link to submit your revised manuscript. Detailed information on submitting your revised paper are below.

Link Not Available

Sincerely,

Lindsey Burbank

Preparing Revision Guidelines

- point-by-point responses to the issues I raised in your cover letter
- Upload a compare copy of the manuscript (without figures) as a "Marked-Up Manuscript" file.
- Each figure must be uploaded as a separate file, and any multipanel figures must be assembled into one file.
- Manuscript: A .DOC version of the revised manuscript
- Figures: Editable, high-resolution, individual figure files are required at revision, TIFF or EPS files are preferred

Please return the manuscript within 60 days; if you cannot complete the modification within this time period, please contact me. If you do not wish to modify the manuscript and prefer to submit it to another journal, please notify me of your decision immediately so that the manuscript may be formally withdrawn from consideration by Microbiology Spectrum.

Dear editor,

Thank you for your thorough review of our work and we are very grateful for your input. We have revised our manuscript thoroughly based on your comments. Details are as follows:

Comment 1: In my assessment of your responses there is just one more point that I would like to see addressed: similar to the points of reviewer #2, I think it is important to make clear that the effect of BR+AM on plant performance in general was quite small although the effect of BR on AM colonization was much more significant. As this was an experimental condition, it is possible that the impact on plant performance would be greater under other environmental conditions and pressures, but that would be an area for additional work. If you could address this point specifically in the conclusions I think it would be beneficial for those who are interested in this area of research and want to build on your results.

Response 1: Thanks for your kind suggestion. According to your suggestion, we have specifically addressed this point in the conclusions. And we have also revised the results and discussions section related to this.

The revised details are as follows:

Line 149-150, To sum up, low concentration of BR and inoculation of AM fungi promoted the growth of *E. grandis* to some extent.

Line 222-225, To a certain extent, BR and AM fungal colonization increased plant fresh weight, promoted the development of plant height and taproot length, inhibited the accumulation of reactive oxygen species, and activated of antioxidant enzymes.

Line 246-249, Although the effect of BR on AM colonization is significant, the joint effects of BR and AM on plant growth can enhance plant growth under certain conditions. It is possible that the impact on plant performance would be greater under other environmental conditions and pressures, but this needs further experiments.

Line 296-300, And the synergistic treatment of BR and AM colonization has quite a small effect on plant

performance, although the effect of BR on AM colonization was much more significant. Moreover, it is possible that their effects on plant performance may be greater under other environmental conditions and stresses, but additional studies are needed to prove this.

All the changes we make in the manuscript are highlighted by coloured text.

If there are any further issues, please don't hesitate to tell us.

Best regards!

Yours

Ming Tang

November 8, 2021

Prof. Ming Tang
South China Agricultural University
College of Forestry and Landscape Architecture
No.483, Wushan Road, Tianhe District
Guangzhou
China

Re: Spectrum01645-21R2 (Brassinosteroids benefit plants performance by augmenting arbuscular mycorrhizal symbiosis)

Dear Prof. Ming Tang:

Your manuscript has been accepted, and I am forwarding it to the ASM Journals Department for publication. You will be notified when your proofs are ready to be viewed.

Sincerely,

Lindsey Burbank
Editor, Microbiology Spectrum
